# Mutational Analysis of Redβ Single Strand Annealing Protein: Roles of the 14 Lysine Residues in DNA Binding and Recombination In Vivo

**DOI:** 10.3390/ijms22147758

**Published:** 2021-07-20

**Authors:** Katerina Zakharova, Brian J. Caldwell, Shalya Ta, Carter T. Wheat, Charles E. Bell

**Affiliations:** 1Department of Biological Chemistry and Pharmacology, The Ohio State University, Columbus, OH 43210, USA; zakharova.4@osu.edu (K.Z.); caldwell.723@osu.edu (B.J.C.); ta.50@osu.edu (S.T.); wheat.35@osu.edu (C.T.W.); 2Ohio State Biochemistry Program, The Ohio State University, Columbus, OH 43210, USA; 3Department of Chemistry and Biochemistry, The Ohio State University, Columbus, OH 43210, USA

**Keywords:** single-strand annealing, DNA recombination, DNA repair, homologous recombination, bacteriophage λ, recombineering

## Abstract

Redβ is a 261 amino acid protein from bacteriophage λ that promotes a single-strand annealing (SSA) reaction for repair of double-stranded DNA (dsDNA) breaks. While there is currently no high-resolution structure available for Redβ, models of its DNA binding domain (residues 1–188) have been proposed based on homology with human Rad52, and a crystal structure of its C-terminal domain (CTD, residues 193-261), which binds to λ exonuclease and *E. coli* single-stranded DNA binding protein (SSB), has been determined. To evaluate these models, the 14 lysine residues of Redβ were mutated to alanine, and the variants tested for recombination in vivo and DNA binding and annealing in vitro. Most of the lysines within the DNA binding domain, including K36, K61, K111, K132, K148, K154, and K172, were found to be critical for DNA binding in vitro and recombination in vivo. By contrast, none of the lysines within the CTD, including K214, K245, K251, K253, and K258 were required for DNA binding in vitro, but two, K214 and K253, were critical for recombination in vivo, likely due to their involvement in binding to SSB. K61 was identified as a residue that is critical for DNA annealing, but not for initial ssDNA binding, suggesting a role in binding to the second strand of DNA incorporated into the complex. The K148A variant, which has previously been shown to be defective in oligomer formation, had the lowest affinity for ssDNA, and was the only variant that was completely non-cooperative, suggesting that ssDNA binding is coupled to oligomerization.

## 1. Introduction

Proteins that bind to single-stranded DNA (ssDNA) and promote the annealing of complementary strands are called single-strand annealing (SSA) proteins [1]. They typically have important roles in DNA recombination, most notably in the SSA pathway for repair of double-stranded DNA (dsDNA) breaks [2], and in the second end capture step of homology directed repair and homologous recombination [3]. At least four different families of SSA proteins have been identified, based on their distinct 3-dimensional folds. These include Rad52 [4,5], the primary SSA protein in eukaryotic cells, bacterial RecO [6], DdrB of *Deinococcus radiodurans* [7], and ICP8 of oncogenic dsDNA viruses such as Herpes Simplex type I [8]. While RecO appears to function as a monomer [6], Rad52 forms oligomeric rings of 7 or 11 subunits [4,5,9], DdrB forms 5-mer rings and face-to-face stacked rings [7,10], and ICP8 forms a variety of structures, including stacked 9-mer double rings [11] and bipolar helical filaments [12].

Crystal structures of the Rad52 N-terminal DNA binding domain in complex with ssDNA [13] reveal two distinct sites on the protein for binding to DNA (Appendix A). One site, referred to as “inner”, sits at the base of a deep and narrow positively charged groove that runs along the upper surface of an 11-mer ring. A 40-nt ssDNA binds to this site with a stoichiometry of four nucleotides per monomer, primarily through interactions involving the sugar-phosphate backbone, leaving the bases exposed for homology recognition (Appendix A). The second site, referred to as “outer”, lies in the same groove, but at its upper rim, as opposed to deep at its base. A shorter segment of ssDNA binds in a helical conformation to this site and bridges two neighboring rings in the crystal lattice, through similar sets of interactions (Appendix A). The authors suggest that this site may have a role in bringing together multiple ring-ssDNA complexes to promote annealing. They also propose a mechanism in which ssDNA is initially bound to the inner site, but then moves to the outer site as the annealing reaction proceeds [13].

Many dsDNA bacteriophages encode an SSA protein of the RecT/Redβ family that shares distant homology with Rad52 [14,15,16]. The current Pfam database (version 32.0) lists 1181 such proteins [17], predominantly encoded within the genomes of bacteriophage, prophage, and mobile genetic elements such as conjugative plasmids [18,19]. Members of this family are often paired with an exonuclease to form a simple two-component “SynExo” recombinase [20,21]. The most extensively studied of these is Redβ from bacteriophage λ, a 261 amino acid protein (*Mr* 29.7 kDa) that is coupled with λ exo, a ring-shaped 5’-3’ exonuclease [22,23,24]. While the exact biological role of these recombination systems has been elusive, recent data from *Vibrio cholerae* suggests that they may have evolved to repair dsDNA breaks formed by CRISPR-Cas systems [18].

Although a high-resolution structure of Redβ, or any member of the RecT/Redβ family is not yet available, the N-terminal DNA-binding domain (residues 1-188) can be modeled based on its distant homology with Rad52 [14,15,16]. Redβ also has a smaller C-terminal domain (CTD, residues 193-261) that binds to λ exo [25], presumably to facilitate loading of Redβ directly onto the 3’-overhang as it is generated by λ exo. A recent crystal structure of the CTD determined in complex with λ exo [26] revealed that it forms a three-helix bundle with a similar fold as phage λ orf, a protein that binds to *E. coli* single-stranded DNA binding protein (SSB) to function as a recombination mediator [27]. Based on this new structural insight, the Redβ CTD was hypothesized (and confirmed) to have a second role in binding to SSB, presumably to gain access to the target site of annealing at the lagging strand of a replication fork [26,28].

There has been considerable interest in understanding the mechanism by which Redβ promotes the SSA reaction, both as a model for understanding SSA proteins in general, and as a powerful tool for bacterial genome engineering [29,30,31]. Along these lines, Redβ exhibits unusual and intriguing DNA binding behavior: it binds weakly to ssDNA substrate, not at all to pre-formed dsDNA, but tightly to a duplex intermediate of annealing formed when two complementary oligonucleotides are added to the protein sequentially [32]. Negative stain EM of Redβ revealed oligomeric structures that closely parallel this DNA binding behavior [33]. The protein forms rings of 10-12 subunits without DNA, larger rings of 15-18 subunits in the presence of ssDNA, and helical filaments in the presence of heat-denatured dsDNA, which presumably is annealed to form a duplex intermediate. Based on these structures, a model for annealing was proposed in which the ring form of the protein binds to ssDNA in a similar manner as seen for the inner site on Rad52. Once a complementary ssDNA is paired and the annealing reaction proceeds, the complex somehow transitions (or reassembles) into a helical filament that binds to the duplex intermediate. Tighter binding of Redβ to the duplex intermediate than to the ssDNA substrate presumably drives the annealing reaction forward [32,33].

While this model is compelling, the individual residues of Redβ that are involved in DNA binding are not resolved in the existing low-resolution EM structures, and difficult to discern from existing homology models, which, though all similar to Rad52, have significant differences in sequence registration (Appendix A). Chemical modification with NHS-biotin identified six lysine residues of Redβ that are protected from modification by DNA binding (K36, K61, K69, K148, K154, and K172), all of which reside within the N-terminal DNA binding domain [34]. Purification of alanine-substituted proteins for three of these (K36A, K69A, and K172A), as well as two negative controls (K11A and K253A), revealed that K172A completely abolishes DNA binding, K36A and K69A weaken DNA binding, and K11A and K253A have little to no effect. However, the mutational analysis in this study tested the activities of these variants for DNA binding in vitro and did not test their effects on DNA recombination in vivo. A separate study probed the effects of mutations at 12 residues of Redβ (including four of the lysines) on DNA binding and oligomerization in vitro, but not in vivo [16].

Here, we mutated all 14 lysine residues of Redβ to alanine and tested their effects on ssDNA recombination in vivo. We also purified the 14 Lys to Ala Redβ protein variants and measured their activities in two different DNA binding assays in vitro, one for binding to ssDNA substrate and another for binding to annealed duplex intermediate. Mapping of the mutational data onto a structural model of the DNA-binding domain and the crystal structure of the CTD provides new insights into Redβ’s mechanism of DNA binding and annealing.

## 2. Results

### 2.1. Mapping of the 14 Lysine Residues onto a Structural Model of Redβ

Although there is no high-resolution structure of full-length Redβ, a crystal structure of the CTD (residues 193 – 261) has been determined [26] and models for the N-terminal DNA-binding domain have been proposed based on distant homology with Rad52 (Figure 1 and Appendix A) [14,15,16]. The CTD contains 5 of the 14 lysine residues (K214, K245, K251, K253, and K258), as shown in Figure 1B. These are not thought to be involved in DNA binding, as the isolated CTD shows no interaction with DNA [26], and the isolated N-terminal domain (constructs as short as residues 1-177) can perform an in vitro annealing reaction on its own [34]. The remaining nine lysines map to the DNA binding domain (K11, K36, K61, K69, K111, K132, K148, K154, K172).

One of the models for the DNA binding domain was made publicly available [15] and is shown in Figure 1A with its seven lysines labeled (only the conserved core of the domain formed by residues 45-188 is shown in the figure). Compared to Rad52, this model has a significantly shorter β3-β4 hairpin and an extended β5-α3 loop, as shown in Appendix A. In Figure 1A, the two sites for binding ssDNA observed in separate Rad52 crystal structures have been mapped onto this model of Redβ by structural superposition. As in Rad52, the “inner” site at the base of the narrow groove has the ssDNA bound in an extended conformation with the bases exposed for homology recognition, while the “outer” site at the upper rim of the groove has the ssDNA bound in a right-handed helical conformation. By analogy with Rad52 [13,35] a mechanism can be proposed in which the first strand of ssDNA binds to the inner site, and then moves up to the outer site as it pairs with the second complimentary strand upon annealing.

Mapping of the lysine residues of Redβ onto this model allows for some predictions regarding their potential roles in DNA binding. The side chains of K154 and K172 are near the inner site, while K69, K132, and K148 are near the outer site. K61 is within 12Å of the inner site but separated from it by the β1-β2 hairpin, such that it would be on the outer surface of the oligomeric ring and facing away from the inner groove (Figure 1C). Meanwhile K111 is on the bottom of the structure, distant from both DNA binding sites, and K11 and K36, which would extend from α1, would likely also be on the bottom of the structure. Based on this model, we can hypothesize that K154 and K172 are involved in binding to the initial ssDNA substrate at the inner site, K69, K132, and K148 are involved in binding to the second strand and/or the annealed duplex intermediate at the outer site, and K11, K36, K61, and K111 are not involved in DNA binding. The following mutational analysis will test this hypothesis.

### 2.2. Roles of the 14 Lysine Residues in ssDNA Recombination In Vivo

The roles of the 14 lysine residues of Redβ in promoting ssDNA recombination in vivo were tested in two different assays, one that uses a pSIM5 vector for temperature-induced expression of Red functions (Redβ, λ exo, and gam) from their native λ P_L_ promoter [36], and a second that uses a pSC101 vector for arabinose-induced expression from a P_BAD_ promoter [37]. We will refer to these assays as “pSIM5” and “pSC101”. Both involve electroporation of an oligonucleotide designed to repair a defective gene into *E. coli* cells induced for Red expression, followed by plating on selective media to score for recombinants. In the pSIM5 assay, successful recombination of a 70-mer oligo repairs a chromosomal copy of the *gal*K gene to enable recombinants to grow on minimal media with galactose as the sole carbon source [36]. In the pSC101 assay, successful recombination of a 100-mer oligo corrects a 4 bp deletion within a neomycin gene encoded on a bacterial artificial chromosome to enable recombinants to grow on kanamycin [26,37]. The two assays have several differences, as detailed in Materials and Methods, and thus serve as independent, parallel tests of the effects of the mutations in Redβ. Expression of a soluble protein of the correct size for the 14 Lys to Ala variants for each plasmid was confirmed by Western blot, as shown in Appendix A. The fact that all 14 variants are expressed in the soluble fraction of the cell lysate suggests that none of the Lys to Ala mutations have a detrimental effect on folding.

The results of the two assays for the 14 Lys to Ala variants are shown in Figure 2 and Appendix A, where the level of recombination is reported as the number of recombinants per 10^8^ viable cells. Each reported value is based on a minimum of six independent measurements for each variant, performed together with WT on at least two different days. For WT Redβ, the pSIM5 assay resulted in 22 ± 31 × 10^5^ recombinants per 10^8^ viable cells (Appendix A), which equates to an efficiency of 2.2%. The pSC101 system resulted in approximately 10-fold lower levels of recombination.

As seen in Figure 2 and Appendix A, most of the mutations decrease the activity of Redβ significantly: 11/14 result in less than 10% of the activity of WT, in both assays. This is not entirely surprising, as each Redβ monomer is potentially engaged in multiple interactions, including with ssDNA substrate, with annealed duplex intermediate, with neighboring Redβ subunits in oligomeric complexes, and with *E. coli* SSB protein [26]. Another observation is that mutations within the putative core DNA binding domain (residues 45-188) tend to have the strongest effects, while mutations that are outside of this domain, either in the N-terminal extension (K11A, K36A) or in the CTD (K214A, K245A, K251A, K253A, and K258A), tend to have lesser effects. A notable exception to this trend is K253A within the CTD, which gave 0.03% of the activity of WT, second lowest in both assays. This can be easily explained, however, as the K253A mutation completely disrupts the interaction with SSB, which is required for recombination in vivo [26]. Another exception is K214A of the CTD, which gave only 0.99% and 0.15% activity in the two assays. K214 is right next to K253 on the surface of the CTD (Figure 1B), such that the K214A mutation could also affect binding to SSB, although this has not been tested.

In general, despite the overall low levels of activity, the results from the two assays are remarkably consistent with one another. This can be seen by the fact that the three variants with the highest activity (K11A, K251A, and K258A) were the same in both assays, as were the two variants with the lowest activity (K61A and K253A). The lone exception to this trend was K154A, which gave 9.2% activity in the pSIM5 assay (5th highest), but only 0.03% activity in the pSC101 assay (3rd lowest).

In summary, the two assays show that the majority of lysines within the core of the DNA binding domain (residues 45-188) are strongly required for in vivo DNA recombination, whereas those outside of the core DNA binding domain are not, with the exception of K253, and to a lesser extent K214, which can be explained by their likely involvement in the interaction with SSB.

### 2.3. Roles of the 14 Lysine Residues in Forming the Complex with Annealed Duplex Intermediate

To determine if the defect in in vivo recombination for any of the Lys to Ala variants was due to a defect in forming the complex with annealed duplex intermediate, each of the 14 variants were purified, and subjected to a gel-based DNA annealing assay with two complementary 50-mer oligonucleotides [14,38]. The proteins were expressed as N-terminally 6xHis-tagged proteins and purified on a small-batch scale on nickel-spin and gel-filtration columns as described in Materials and Methods. All of the proteins were present in normal amounts in the soluble fraction of the cell lysate, suggesting that none of the mutations have a detrimental effect on folding. SDS-PAGE analysis of the purified proteins is shown in Appendix A. The activity of WT Redβ purified in this manner was indistinguishable from that of the protein purified by regular full-scale column chromatography [25] (data not shown).

As shown in Figure 3, the DNA binding assay involves two complementary 50-mer oligonucleotides, labeled at the 5’-end with either Cy3 (green) or Cy5 (red). The 10 μM Redβ was mixed with a slight excess (5 nt per monomer) of each oligonucleotide individually (lanes labeled “ss”), or with the two oligos added sequentially (lanes labeled “ad” for annealed duplex), where one oligo was incubated with the protein for 30 min before the second oligo was added. Complexes were then analyzed by agarose gel electrophoresis. With either oligo individually, WT Redβ formed a faint shifted band with significant streaking, indicative of a weak complex. When incubated with the two oligos added sequentially, however, WT Redβ formed a complex that was seen as a much more prominent band of slightly higher mobility, with no streaking. The yellow color of this band indicates that the complex contains both oligonucleotides, presumably paired to one another to form a duplex intermediate of annealing. This was confirmed by exposures from the individual 550 nm (Cy3) or 650 nm (Cy5) channels, as shown in Appendix A. In Appendix A, control reactions show that the same complex is formed regardless of which complementary strand is added first (lanes labeled “ad cy3” and “ad cy5”), that the complex with both strands is not formed when the two strands are non-complementary to one another (lanes labeled “NC cy3” and “NC cy5”), and that no complex is formed when the two complementary strands are annealed to one another prior to addition of Redβ (lanes labeled “PAD”).

As seen in Figure 3, six of the Lys to Ala variants (K61A, K111A, K132A, K148A, K154A, and K172A) showed no interaction with the oligonucleotides, either as ssDNA or as annealed duplex, indicating that they are completely defective in DNA binding. All of these variants involve lysines located within the core of the DNA binding domain (residues 45-188), and all showed low levels of recombination in vivo, ranging from 0% to 9% of WT. The remaining mutants were able to form the complex with annealed duplex, as indicated by a distinct band of the same color and migration as seen for WT. With the exception of K69A, all of these mutants are in regions of the protein that lie outside of the core DNA binding domain, either in the N-terminal extension (K11A and K36A) or in the CTD (K214A, K245A, K251A, K253A, and K258A).

K69A is unique in that it showed significantly enhanced interaction with ssDNA, while also forming the complex with annealed duplex. Interestingly, this mutant showed low levels of recombination in vivo (2.5–2.8% of WT), possibly due to an imbalance in relative affinity for ssDNA substrate vs. annealed duplex product. K253A is unique in that it formed a significant amount of annealed duplex product in vitro, but almost no recombination activity in vivo. However, this can again be explained by the fact that K253 is required for the interaction with SSB, which is in turn required for recombination in vivo [26].

In summary, there is in general close agreement between the level of recombination observed in vivo, and the ability of each variant to form the complex with annealed duplex in vitro (Table 1). Two exceptions to this trend are K69A, which has low activity in vivo, but increased binding to ssDNA in vitro, and K253A, which is defective for recombination in vivo due to its loss of interaction with SSB, despite retaining close to normal DNA binding and annealing activities in vitro.

### 2.4. Roles of the 14 Lysines in Binding to ssDNA Substrate

It is conceivable that a given lysine residue of Redβ is important for forming the complex with annealed duplex, but not for binding to the first strand of ssDNA. While the gel-based assay presented above tested the abilities of the 14 Lys to Ala variants of Redβ to bind to 50-mer ssDNA and to 50-mer annealed duplex, little binding to ssDNA was observed, even for WT, likely due to dissociation of the weaker ssDNA complex during electrophoresis [32]. Therefore, to quantitatively assess the binding of the 14 variants to ssDNA substrate, fluorescence anisotropy titrations were performed, using a 5’-fluorescein labeled 48-mer oligonucleotide. As described previously [25], these assays used a low-salt buffer instead of the PBS buffer used above, to maximize binding and therefore better differentiate between the mutants. Three independent titrations were performed for WT and each variant, and the data were fit to the Hill equation to determine the apparent *K_d_* values, which are reported as mean and standard deviation in Table 1. Representative curves for each mutant are shown in Figure 4, plotted in groups of 3–4 with WT, in order of their positions in the amino acid sequence.

The results of the ssDNA-binding assay closely parallel what was seen for annealed duplex. Mutations at residues within the C-terminal domain (K214A, K245A, K251A, K253A, and K258A) have minimal effect on ssDNA binding (at most a 2.8-fold increased *K_d_* for K214A), while mutations at almost all of the lysines within the DNA binding domain affected ssDNA binding much more severely. The weakest binders were K148A (331-fold increased *K_d_*), K172A (24-fold), K111A (22-fold), K154A (19-fold), and K132A (16-fold). Residues at the N-terminal portion of the DNA-binding domain had more moderate effects: 2.1-fold increased *K_d_* for K11A, 4.8-fold for K36A, 2.9-fold for K61A, and 3.7-fold for K69A. Most of the variants had reduced cooperativity of ssDNA binding, as seen from the Hill coefficients (*n*) (Table 1), which were all lower than WT (though not significantly so for some). K148A had a significantly lower Hill coefficient (0.87 ± 0.1) than all other variants, consistent with prior observations that it is defective in oligomer formation [16].

K61A stands out as being the only variant that did not form the complex with annealed duplex at all, but still had close to normal ssDNA-binding (only 2.9-fold increased *K_d_*). This points to a specific role for K61 in binding to the complex with annealed duplex. Several other variants showed decreased binding to annealed duplex (less intense band for the complex), but close to normal ssDNA binding (*K_d_* within 2-fold of WT). However, as the duplex binding assay was not quantitative, the specific roles of these residues are less clear. Additionally of note, the K69A variant that showed increased binding to ssDNA in the gel-based DNA-binding assay, did not show increased affinity for ssDNA by fluorescence anisotropy, possibly due to the different buffer conditions.

In summary, the mutations that have strong effects on forming the complex with annealed duplex product also tend to have strong effects on ssDNA-binding, making it difficult to dissect out specific roles of particular residues. The one exception to this trend is K61A, which showed no binding to annealed duplex, but only 2.9-fold decreased affinity for ssDNA. In general, mutations within the CTD have only moderate effects on ssDNA-binding, while the mutations within the core DNA-binding domain have more significant effects.

## 3. Discussion

Low resolution EM images of Redβ have led to a compelling model in which the protein binds to initial ssDNA substrate as an oligomeric ring, and transitions (or reassembles) to a helical filament when a second strand of complementary ssDNA is added to form the complex with annealed duplex [32,33]. This model inspires intriguing questions, such as, to what extent do the residues of Redβ that contact ssDNA in the ring complex overlap with those that contact annealed duplex in the helical filament? In the absence of higher resolution structures of these complexes, however, it has been difficult to discern the roles of individual residues of Redβ in DNA binding and annealing. It is possible to construct models of the DNA binding domain of Redβ based on its distant homology with Rad52 [14,15,16] and further to map out putative DNA-binding sites based on Rad52 co-crystal structures [13]. However, due to the low level of sequence identity between the two proteins, the registrations of the sequence alignments are far from certain in several places, which in turn makes positioning of key residues of Redβ relative to the two segments of ssDNA ambiguous.

To test these models, we focused on lysine residues, because lysine is positively charged and commonly involved in DNA binding, and because of our prior study that used chemical modification with NHS-biotin to identify 6/14 lysines as being protected from modification in the complex with annealed duplex [34]. However, this prior study did not test the effects of the mutations on forming the complex with ssDNA substrate, or on DNA recombination in vivo. Thus, to extend our prior analysis, particularly in light of the new homology models, we have now assessed the effects of all 14 Lys to Ala mutations on ssDNA recombination in vivo, and on binding to both ssDNA substrate and annealed duplex product in vitro.

At the domain level, the results of our analysis clearly support the separation of function between the N- and C-terminal domains. None of the mutations within the CTD strongly affected DNA binding, either to ssDNA substrate or to annealed duplex product. Two of the CTD mutations, K253A and K214A, did have strong effects on in vivo recombination, but this was almost certainly due to their effects on the interaction with SSB: the K253A mutation has previously been shown to abolish the interaction with SSB in vitro [26], and the side chains of K253 and K214 are right next to each other on the surface of the CTD, suggesting K214 is also likely to be at the Redβ-SSB interface. By contrast, most of the mutations within the core of the DNA-binding domain (residues 45-188) had strong effects on DNA binding. K69A was the only variant within this core domain (out of seven total) that could form the complex with annealed duplex, and K61A and K69A were the only variants that could bind to ssDNA with close to normal affinity. Collectively, the mutations within the DNA-binding domain also tended to have the strongest effects on DNA recombination in vivo. Finally, the two variants at the most N-terminal region of the protein, K11A and K36A, were able to form the complex with annealed duplex, had only mildly reduced affinity for ssDNA, and retained at least some activity in vivo (1.7–17% of WT).

One of our goals in this study was to identify residues of Redβ that are uniquely defective for binding to annealed duplex intermediate, to test our hypothesis for the two different DNA binding sites mapped out by comparison with Rad52 (Figure 1A). As binding to the first ssDNA is a pre-requisite for forming the complex with both strands, we did not expect to observe variants that are uniquely defective for binding to ssDNA, which was indeed the case. We did however find one variant, K61A, which was only mildly impaired for binding to ssDNA, but completely defective for binding to annealed duplex. Unexpectedly, however, this residue did not map to either of the DNA-binding sites in either of the models (Figure 1A). In general, mutations at many of the residues that are not close to either DNA binding site in the model, including K36A, K61A, and K111A, had significant effects on DNA binding and in vivo recombination. The K111A variant is particularly striking, as it is located on the complete opposite side of the structure from the putative DNA-binding site (Figure 1). Thus, while the model is accurate enough to rationalize the DNA binding data at the domain level, it is apparently not accurate enough to rationalize the data at the level of individual residues.

In a previous study, Matsubara et al. examined the effects of mutations at 12 amino acids of Redβ on DNA binding and oligomerization in vitro [16]. As four of their mutations were at lysine residues (K69A, K132A, K148A, and K172A), the results for these variants can be compared. Interestingly, K69A stood out as being unusual in both studies. In Matsubara et al., K69A showed only moderately impaired binding to ssDNA and annealed duplex but had significantly altered kinetics of annealing: two complementary 50-mer oligos were seen to anneal to 50% completion in 7 min without Redβ, in less than 1 min with WT Redβ, but in 15 min with the K69A variant. K69A not only slowed the rate of annealing but also prohibited the reaction from proceeding to completion. In our study, K69A was remarkable in having increased binding to ssDNA substrate in the gel shift assay, which could conceivably alter the balance needed for efficient catalysis of annealing. This effect manifested as a 40-fold reduction in recombination levels in vivo, to 2.5–2.8% of WT.

For the other three variants that were tested in both studies, K132A, K148A, and K172A, our data tend to show stronger effects of the mutations on DNA binding, both to ssDNA substrate and to annealed duplex product. For example, we observed no complex with annealed duplex for all three of these variants, whereas Matsubara et al. observed appreciable amounts of this complex for K132A and K172A (though none for K148A). Similarly, in our study K148A and K172A were the 1st and 3rd most defective variants in binding to ssDNA, whereas in Matsubara et al., all three of these variants showed closer to normal binding to ssDNA. These differences can likely be attributed to several experimental differences. For example, Matsubara et al. assessed ssDNA binding by gel-shift after crosslinking of protein–DNA complexes with 0.1% glutaraldehyde, whereas we performed an equilibrium measurement using fluorescence polarization. Our method, which did not employ crosslinking, is likely to better differentiate the variants with regard to binding affinity. Our experiment was limited to 10 μM Redβ, whereas Matsubara et al. tested Redβ concentrations as high as 100 μM. Although prior studies indicated that the concentration of Redβ in cells active for recombination is lower than 150 nM [39], we have recently re-examined the in vivo concentration of Redβ in two different expression systems commonly used for recombineering and found it to range from 7 to 27 μM [40]. With regard to DNA annealing, both studies used 50-mer oligonucleotides to assess the formation of the complex with annealed duplex. Our study used a single concentration of Redβ (10 μM) in PBS buffer at physiological ionic strength, while Matsubara et al. performed a titration with 0.1–2.7 μM Redβ in low salt buffer (20 mM Tris pH 7.5, 5 mM MgCl_2_, 1 mM DTT, 0.1 mg/mL BSA). We used a single concentration of Redβ instead of a titration as due to the sequential addition of complementary oligonucleotides, as well as the lack of binding of Redβ to pre-formed dsDNA, this experiment is not an equilibrium measurement, and thus a meaningful binding affinity cannot be obtained. Furthermore, we have recently demonstrated that formation of the complex with annealed duplex is not sensitive to ionic strength or buffer composition and does not require added Mg^2+^ ion [40]. Nonetheless, we consider measurements at physiological ionic strength to be most meaningful.

A notable insight from Matsubara et al. was that the K148A (and R149A) protein was defective in oligomerization, as seen by native PAGE, gel filtration, and negative stain EM [16]. This result is of particular interest because there has been some debate as to whether or not the oligomeric rings seen for WT Redβ are the active species for annealing in vivo [14,39]. In our study, K148A did not form the complex with annealed duplex, had the weakest binding of all variants to ssDNA (330-fold reduced compared to WT), and 0.26 or 1.3% of the in vivo activity of WT. Moreover, K148A was the only variant for which the ssDNA-binding was completely non-cooperative (Hill coefficient = 0.87 ± 0.1). These data would tend to suggest that the defect in oligomerization is likely to be responsible for the defects in DNA binding and recombination. In Matsubara et al., K148A had moderately reduced binding to ssDNA, severely weakened binding to annealed duplex, and moderately reduced kinetics of annealing. Interestingly, by EM, K148A showed no ring formation in the absence of DNA but formed partial rings and filaments in the presence of ssDNA. All together, these data point to significant defects in the activity of K148A, supporting the notion that oligomerization is coupled to annealing, at least at some stages of the reaction.

In summary, our results on the activities of the 14 Lys to Ala variants show that the residues involved in DNA binding, both to ssDNA substrate and to annealed duplex product, reside predominantly if not exclusively within the DNA binding domain, consistent with prior conclusions that the CTD is not involved in DNA-binding [25]. Lys-11 and Lys-36 are also not critical for DNA binding or in vivo activity, and likely reside on an N-terminal extension of the DNA binding domain, as predicted in the models. K61 is identified as being important for binding to annealed duplex, but not to ssDNA substrate, providing new information towards mapping out the different DNA-binding sites on the protein. The K148A variant, which was previously shown to be defective in oligomerization, is shown here to be defective in DNA binding in vitro and recombination in vivo, suggesting that oligomerization is important for annealing. Lysines 214 and 253, which are close together on the surface of the CTD, were identified as being important for recombination in vivo, but not for DNA binding in vitro, re-enforcing their likely roles in binding to SSB, as well as the requirement of this interaction for recombination in vivo.

## 4. Materials and Methods

### 4.1. In Vivo DNA Recombination Assays

Two different ssDNA recombination assays, also called “oligo repair” or “ssOR”, were performed to measure the in vivo activities of the 14 Lys to Ala variants. We will refer to these as the “pSIM5” and “pSC101” assays, based on the plasmids used for expression of Red functions. In the pSIM5 assay, Red functions (*redα*, *redβ*, and *gam*) were expressed from a pSIM5 plasmid from the native λ P_L_ promoter under the control of a temperature sensitive λ repressor variant cI857 [36]. In the pSC101 assay, Red functions were expressed from a pSC101-BAD-gbaA-tet plasmid from a P_BAD_ promoter under the control of arabinose [37]. The 14 Lys to Ala variants were introduced into each of these plasmids by the QuikChange^TM^ PCR method (Agilent Technologies, Santa Clara CA, USA). The resulting 28 PCR products were digested with *Dpn*I, transformed into chemically competent DH5-alpha *E. coli* cells, and selected on LB-agar containing 30 μg/mL chloramphenicol (Cm30, pSIM5) or 4 μg/mL tetracycline (Tet4, pSC101). Single colonies were grown in 5 mL overnight LB cultures (with Cm30 or Tet4) and plasmids were isolated using the QIAprep Spin Miniprep Kit (Qiagen, Germantown MD, USA) and confirmed by Sanger dideoxy DNA sequencing (Ohio State Comprehensive Cancer Genomics Facility, Columbus OH, USA). The pSC101 ssDNA recombination assay was performed in GB2005 cells as described previously [26].

For the pSIM5 assay, pSIM5 plasmids expressing WT or the 14 Lys to Ala mutants were transformed into electro-competent HME57 *E. coli* cells [36] and selected on LB-agar with Cm30. HME57 cells contain a defective chromosomal copy of the *galK* gene, and do not grow on minimal media with galactose. The 5 mL overnight cultures of HME57-pSIM5 were grown from single colonies in LB-Cm30 at 32 °C. The following morning, 0.5 mL of each overnight culture was added to 35 mL of fresh LB-Cm30 in a 125 mL baffled flask and shaken in a water bath shaker at 32 °C for two hours. Then, 15 mL of each culture was transferred to a new 50 mL Erlenmeyer flask, and shaken for 15 min at 42 °C in a separate water bath shaker to induce expression of Red functions. Flasks were then placed on ice for 15 min, after which cells were collected by centrifugation at 6500× *g* for 7 min at 4 °C. After removal of the supernatant, cells were gently re-suspended in 1 mL of ice-cold ddH_2_O, and an additional 30 mL of ice-cold ddH_2_O was added. After centrifugation and removal of the second supernatant, cells were re-suspended in 1 mL ice-cold ddH_2_O, and transferred to a sterile pre-chilled 1.5 mL microcentrifuge tube. After centrifugation at 13,000 rpm in a refrigerated Eppendorf microcentrifuge, the supernatant was removed, cells were re-suspended in 200 μL of ice-cold ddH_2_O, and 50 μL aliquots were transferred to each of three new pre-chilled 1.5 mL microcentrifuge tubes. In total, 2 μL of 5 pmol/μL repair oligo #144 [36] (HPLC-purified, Integrated DNA Technologies, Coralville IA, USA) was added to each tube, and the mixture was transferred to a pre-chilled 0.1 cm gap electroporation cuvette (product # 9104-1050, USA Scientific Ocala FL, USA). Cells were electroporated at 1.8V using a Gene Pulser (Bio-Rad, Hercules CA, USA), transferred with 950 μL of LB back to the microcentrifuge tube, and shaken for 70 min at 32 °C at 1000 rpm in an Eppendorf thermomixer. Cells were then diluted in LB in 10-fold serial increments to the appropriate levels for counting colonies, and 200 μL was plated on minimal media with 0.2% of D-galactose to count the number of recombinants (plates incubated 2–3 days at 32 °C), and on LB-agar to count the number of viable cells surviving electroporation (plates incubated 1 day at 32 °C).

Colonies for both assays (pSIM5 and pSC101) were counted manually, and the level of recombination reported as the number of colonies per 10^8^ viable cells. The mean (±SD) values reported in Appendix A for each Lys to Ala variant are based on a minimum of 6 independent measurements on at least two separate days (in groups of three). Outlier values were excluded, as determined by a Grubb’s test (Extreme Studentized Deviate method) in GraphPad Prism (San Diego, CA, USA) with a significance level (alpha) of 0.05. There were 7 such outliers for the pSIM5 assay (out of 165 independent measurements total), and 9 outliers for the pSC101 assay (out of 138 independent measurements total). Data from each assay were plotted with error bars representing the standard error of the mean (SEM), and assessed for statistical significance (relative to WT) by an unpaired two-sided Student’s T-test with Welch’s correction for unequal variances and sample sizes (* *p* < 0.01, ** *p* < 0.001, *** *p* < 0.0001).

### 4.2. Western Blots

The expression levels of the 14 Lys to Ala Redβ variants from the pSIM5 and pSC101 plasmids were evaluated by Western blot using an anti-Redαβ antibody generously provided by Dr. Kenan Murphy (UMASS Medical School, Worcester MA, USA). Cell lysates were prepared from HME57 (pSIM5) or GB2005 (pSC101) *E. coli* cell cultures grown and induced under the same conditions used for the ssDNA recombination experiments, except that the culture volume was increased to 50 mL to obtain enough cells for sonication and isolation of the soluble fraction, to ensure that each variant was folded and soluble. Immediately after induction, the OD^600^ was measured, cells were collected by centrifugation, re-suspended in 2.5 mL of buffer containing 50 mM NaH_2_PO_4_, 500 mM NaCl, 10 mM imidazole, 10% glycerol, pH 8.0, and frozen at −80 °C. Frozen cell suspensions were thawed, lysed by sonication, and centrifuged at 15,700× *g* for 30 min. The supernatant containing 2.5 mL of each soluble lysate was stored in 0.5 mL aliquots at −80 °C. Gel samples were prepared by mixing 15 μL of each thawed lysate with 5 μL of 2.5× SDS-PAGE loading buffer, 17 μL of which was loaded onto a 13.5% SDS-PAGE gel. After electrophoresis at 22 °C, bands were transferred to nitrocellulose membranes (Bio-Rad) in transfer buffer (25 mM Tris pH 8.3, 192 mM Glycine, 0.05% SDS, 20% methanol) for 70 min at 100 V (constant voltage) at 4 °C. Membranes were blocked with Fluorescent Blocker (MilliporeSigma, Darmstadt, Germany) for 1.5 h at 22–25 °C, and incubated with anti-Redαβ primary antibody and mouse anti-GAPDH primary monoclonal antibody (Thermo Fisher Scientific, Waltham MA, USA) in Fluorescent Blocker overnight at 4 °C. Membranes were then washed three times for 15 min in PBS with 0.1% Tween^®^ 20, and incubated with IRDye 680RD Donkey anti-Rabbit secondary antibody (Li-COR, Lincoln NE, USA) and IRDye 800 donkey anti-mouse secondary antibody (Li-COR) in Fluorescent Blocker for 45 min at 22–25 °C. Membranes were then washed for 3 × 15 min in PBS, 0.1% Tween^®^ 20, and visualized using an Odyssey Blot Imager (LI-COR, Lincoln NE, USA) with scanning at 700 and 800 nm.

### 4.3. Expression and Purification of the 14 Lysine to Alanine Redβ Variants

Redβ protein used for DNA binding assays was expressed from a pET-28b vector, with the gene for Redβ (UniProt Accession ID A0A1M2R4R6) cloned between the *Nde*I and *Bam*HI restriction sites to produce an N-terminally 6xHis tagged protein with an intervening site for thrombin cleavage [25,34]. Plasmids for expression of the 14 Lys to Ala variants were constructed using the QuikChange^TM^ procedure and confirmed by Sanger dideoxy DNA sequencing. Plasmids were transformed into chemically competent BL21-AI cells, and selected on LB-agar with 50 μg/mL kanamycin. Single colonies were grown in 5 mL overnight cultures, diluted the following morning at 1:100 in 50 mL LB, grown for 2 h at 37 °C to an OD^600^ of ~0.5, induced with 0.2% arabinose and 1 mM IPTG, and shaken at 37 °C for three hours post-induction. Cells were collected by centrifugation for 10 min at 10,000× *g*, re-suspended in 5 mL of Sonication Buffer (50 mM NaH_2_PO_4_, 300 mM NaCl, 10 mM imidazole, pH 8.0), and frozen at −80 °C. Each 5 mL cell suspension was thawed and incubated for 1 h on ice with 1 mg/mL lysozyme, 1 μg/mL leupeptin and pepstatin, and 1 mM PMSF. Cells were lysed using a Branson sonicator with microtip at 30% duty, 70% power, for 2 × 2 min, and clarified by centrifugation twice at 38,000× *g* for 30 min. Clarified lysates were loaded onto Ni-NTA microcentrifuge spin columns (Qiagen, Germantown MD, USA) that had been equilibrated with Sonication Buffer. After centrifugation, the loaded columns were washed four times with 500 μL of 30 mM imidazole in Sonication Buffer and eluted four times with 625 μL of 500 mM imidazole in Sonication Buffer, to give a final volume of 2.5 mL. For each individual protein variant (WT or mutant), this procedure was performed twice to yield a total volume of 5 mL. Samples were then loaded in 2.5 mL increments onto a PD-10 desalting column (product # 17085101, Cytiva, Marlborough MA, USA) equilibrated with PBS, and eluted twice with 3.5 mL of PBS. Each protein was then concentrated to 500 μL (2-4 mg/mL) using an Amicon Ultra 2 mL MWCO 10 kDa centrifugal filter (product # UFC203024, MilliporeSigma, Darmstadt, Germany). Three units of thrombin protease (product # T9326, MilliporeSigma, Darmstadt, Germany) were added to each protein to remove the N-terminal 6xHis tag and incubated at 22 °C for 12-15 h. The resulting purified proteins were stored at −80 °C in 100 μL aliquots. The proteins retain an extra N-terminal Gly-Ser-His sequence after thrombin cleavage, which does not affect the activity of WT Redβ based on several assays in vitro and in vivo [25]. Protein concentrations were determined by OD at 280 nm using an extinction coefficient of 34,950 M^−1^ cm^−1^, which was calculated from the amino acid sequence of WT Redβ, which has five tryptophan residues.

### 4.4. Gel-Based Assay for Binding to Annealed Duplex Product

The gel-based DNA binding assay followed a previously reported method using two complementary 50-mer oligonucleotides, one labeled with 5’-Cy5 (50+) and the other labeled with 5’-Cy5 (50-) [38]. Each 50 μL binding reaction contained 10 μM purified Redβ (WT or Lys to Ala variant) in PBS. For the complexes with annealed duplex, the 50+ oligo (50 μM nucleotides) was added to each protein and incubated for 30 min at 37 °C, after which an equivalent amount of the 50- oligo was added, and incubated for an additional 30 min. For the complexes with ssDNA, only one oligo (either 50+ or 50-) was added, and incubated for 30 min at 37 °C. After addition of 3 μL of 10x Orange G Loading Dye (65% (*w*/*v*) sucrose, 20 mM Tris pH 7.5, 250 mM EDTA, 0.3% (*w*/*v*) Orange G dye), 5 μL of each sample was loaded onto a 1% agarose gel, and electrophoresed at 90 V for 40 min at 22 °C. Gels were directly exposed (without drying) using the 550 nm (Cy3) and 650 nm (Cy5) channels of a Sapphire Biomolecular Imager (Azure Biosystems).

### 4.5. Fluorescence Anisotropy Assay for Binding to ssDNA Substrate

To quantitatively compare the binding of the 14 Lys to Ala variants of Redβ to ssDNA, fluorescence anisotropy titrations were performed using a 48-mer 5’-fluorescein labeled oligonucleotide (FAM48, Integrated DNA Technologies, Coralville IA, USA), as described previously [25]. Briefly, 10 nM (molecules) of the FAM48 ssDNA was incubated with varying concentrations (18–10,000 nM monomer) of WT or mutant Redβ in Binding Buffer (50 mM Tris-HCl pH 7.5, 1 mM dithiothreitol, and 0.1 mg/mL BSA). Each 70 μL binding reaction was prepared by adding 63 μL of 1.1x Binding Buffer with FAM48, to 7 μL of 0.75× serially diluted protein (at 10×), to give 24 concentrations total, including one with no Redβ. After equilibration for 60 min at 23 °C, 3 × 20 μL of each 70 μL reaction was pipetted into a Corning 384 well black polystyrene plate, and fluorescence anisotropy was measured for each 20 µL sample in triplicate at 490 nm excitation, 515 nm emission and 23 °C using a Spectra Max M5 Microplate Reader (Molecular Devices, San Jose CA, USA). To determine the apparent dissociation constant (*K_d_*) for each mutant, mean fluorescence anisotropy was plotted against the concentration of Redβ monomer and fit to the following (Hill) equation using KaleidaGraph version 4.0 (Synergy Software, Reading PA, USA):*A* = *A_min_* + (*A_max_* − *A_min_*) × [1/(1+(*K_d_*/*A_min_*)*^n^*],
where *A* is the mean anisotropy (from triplicate measurements at each titration point), *A_min_* is the fitted minimum anisotropy, *A_max_* is the fitted maximum anisotropy, and *n* is the Hill coefficient. Three such titrations were performed for each mutant (except for K61A where only two were performed) to determine values for the mean and standard deviation for the *K_d_* of each mutant.

## Figures and Tables

**Figure 1 ijms-22-07758-f001:**
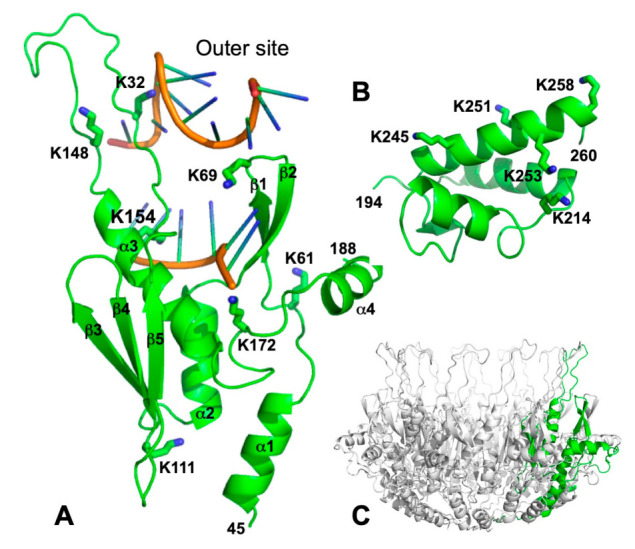
Structural model of Redβ and mapping of possible DNA binding sites. (**A**) Model of the core of the DNA binding domain of Redβ (residues 45-188) based on homology with Rad52 [15]. Seven lysine residues are shown in stick form and labeled. The backbone conformation around K172 has been adjusted to more closely match the structure of Rad52, which has a glycine at the equivalent position. Two segments of ssDNA, aligned from structures of complexes with the Rad52 DNA-binding domain [13], are shown with their sugar-phosphate backbones in orange. The ssDNA at the “inner” site in the middle of the figure is from PDB 5XRZ. Notice that this ssDNA is bound in an extended conformation with its bases exposed for homology recognition. This ssDNA is present in the crystal as a dT40 oligonucleotide that wraps around an 11-mer ring with 4 nt/monomer (as shown in Appendix A). The ssDNA at the “outer” site is from PDB 5XS0. Notice that it is bound in a right-handed helical conformation. The two ssDNAs were not bound simultaneously to Rad52 (as depicted here for Redβ) but rather were taken from separate crystal structures [13]. (**B**) Structure of the Redβ CTD with five lysine residues labeled. The coordinates are from PDB 6M9K [26]. The orientation relative to the NTD in panel A is hypothetical, but the two are positioned with nearest connecting residues juxtaposed. (**C**) Model of the Redβ DNA binding domain in an 11-mer ring, based on Rad52 [15]. The subunit highlighted in green is in a similar orientation as that in panel A.

**Figure 2 ijms-22-07758-f002:**
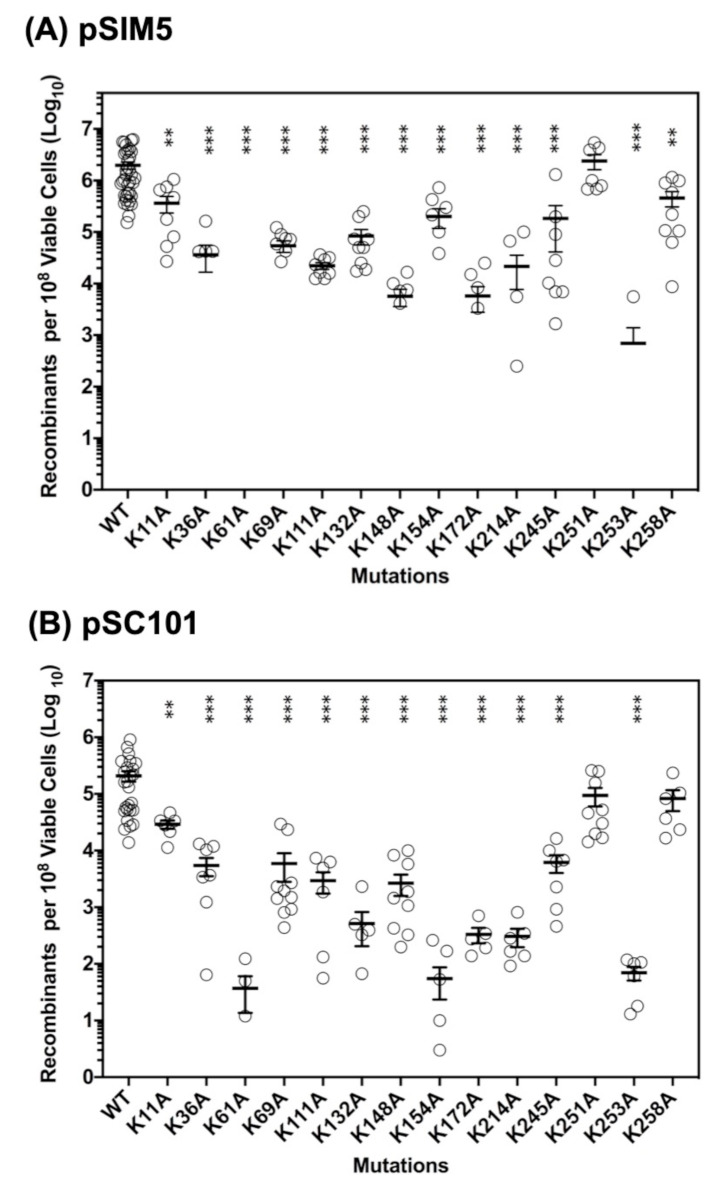
DNA recombination levels for the 14 Lys to Ala mutants. (**A**) Assay using pSIM5 expression vector. (**B**) Assay using pSC101 expression vector. In both cases a repair oligonucleotide is electroporated into cells expressing Red proteins (including Redβ), and after recovery the cells are plated on selective media to score for recombinants, as described in Materials and Methods. The error bars show the mean +/− SEM from a minimum of six independent measurements. Individual experiments giving no colonies (Log_10_(0) = −infinity) are not plotted, including 6/6 for K61A-pSIM5, 5/6 for K253A-pSIM5, 3/6 for K61A-pSC101, and a few others. Statistical significance (relative to WT) was assessed by unpaired two-sided Student’s T-test with Welch’s correction (** *p* < 0.001, *** *p* < 0.0001).

**Figure 3 ijms-22-07758-f003:**
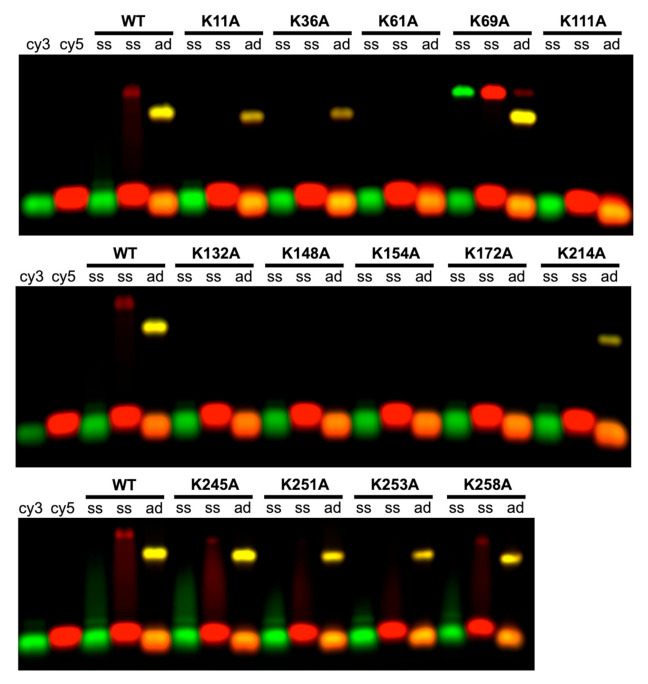
Gel-based DNA binding and annealing assay for the 14 Lys to Ala variants. Lanes labeled “cy3” and “cy5” contain individual 50-mer oligonucleotides labeled at the 5’-end with either Cy3 or Cy5. Lanes labeled “ss” contain 10 μM of Redβ mixed with 50 μM (nucleotides) of each individual nucleotide (Cy3 or Cy5). Lanes labeled “ad” contain Redβ incubated with the two complementary oligonucleotides added sequentially to form the complex with annealed duplex. Notice that WT Redβ interacts weakly if at all with each individual ssDNA (seen as faint bands with streaking) but forms a stable complex when the two complementary oligos are added to the protein sequentially (prominent yellow bands). Many of the Lys to Ala variants, however, do not form this complex. K69A is unique in having increased binding to ssDNA. Single channel exposures for these gels showing each individual oligonucleotide (Cy3 or Cy5) are presented in Appendix A, to confirm that the yellow bands for the complex contain both oligonucleotides.

**Figure 4 ijms-22-07758-f004:**
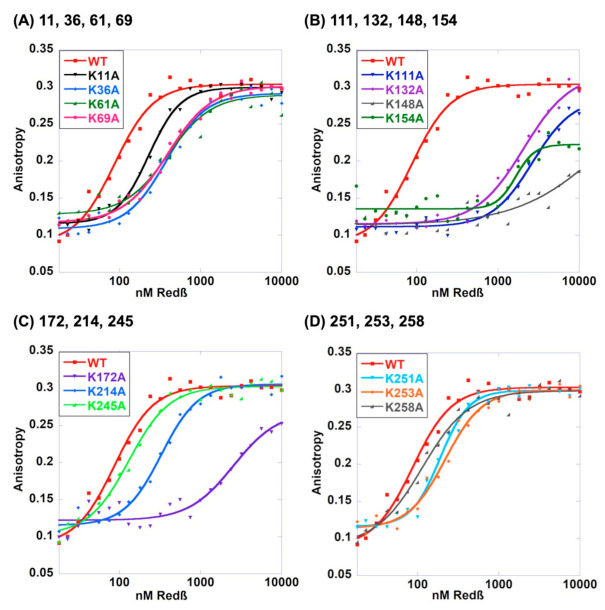
Binding of the 14 Lys to Ala Redβ variants to 5’-FAM 48-mer ssDNA. Groups of three or four variants are plotted together with the same data for WT in panels (**A**)–(**D**). Each data point shows the average anisotropy value from three replicates at each titration point. The solid lines represent the fit of the averaged data points at all concentrations for each titration to the Hill equation, as described in Materials and Methods. The apparent *K_d_* values reported in Table 1 are the mean ± SD values from three independent titrations, one of which, that giving the middle value, is shown here.

**Table 1 ijms-22-07758-t001:** Summary of DNA binding and in vivo recombination activity for each variant.

	In Vitro DNA Binding			In Vivo Recombination (% of WT)	
Redβ Variant	ssDNA ^1^		Annealed Duplex ^2^	pSIM5	pSC101
	(*K**_d_*_,_ nM)	(*n*)			
WT	110 ± 70	2.5 ± 0.9	+	100	100
K11A	230 ± 40	2.2 ± 0.1	+	17	14
K36A	530 ± 310	1.6 ± 0.2	+	1.7	2.6
K61A	390 ± 10	1.6 ± 0.01	-	0	0.02
K69A	410 ± 150	1.4 ± 0.4	+	2.5	2.8
K111A	2400 ± 560	1.6 ± 0.4	-	1.0	1.4
K132A	1770 ± 600	1.3 ± 0.2	-	3.9	0.43
K148A	14,100 ± 14,500	0.87 ± 0.1	-	0.26	1.3
K154A	2100 ± 1100	1.8 ± 1.5	-	9.2	0.03
K172A	2600 ± 240	1.4 ± 0.2	-	0.26	0.16
K214A	310 ± 90	1.7 ± 0.2	+	0.99	0.15
K245A	160 ± 40	1.7± 0.2	+	9.5	3.0
K251A	190 ± 30	2.3 ± 0.4	+	110	60
K253A	210 ± 30	1.8 ± 0.2	+	0.03	0.03
K258A	110 ± 40	1.9 ± 0.4	+	21	40

^1^ For binding to ssDNA (2nd column), the reported values for the apparent dissociation constant (*K_d_*) and Hill coefficient (*n*) are the mean ± SD determined from fits of three independent fluorescence anisotropy titrations to the Hill equation, a representative of which is shown in Figure 4 for each mutant. ^2^ The + or − symbols in the “annealed duplex” column indicate if a complex with annealed duplex was formed, as seen from the yellow band for each variant in the gel of Figure 3.

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
