# Peer review of "Mutational Analysis of Redβ Single Strand Annealing Protein: Roles of the 14 Lysine Residues in DNA Binding and Recombination In Vivo"

_ijms, 2021, doi:10.3390/ijms22147758_

Round 1

Reviewer 1 Report

Zakharova  et al present an article on the RedB Protein and the roles of the 14 lysine residues in DNA binding and recombination in vivo .

The article is interesting and well structured.

However, I have some doubts regarding the results published by Matsubara in 2013 (doi: 10.1371 / journal.pone.0078869) which contrast with those present in this work. Although the authors state that the differences may be dependent on experimental conditions, these may make the interpretation of the data unclear.

Could the authors carry out an experiment under the same conditions as Matsubara to validate their claim and the experimental differences obtained?

Author Response

The reviewer raises a valid point regarding differences between some of our results and those of Matsubara et al. Since the editor Dr. Qui requested our re-submission within three business days, we do not have enough time to perform the additional experiments under the conditions of Matsubara et al., which would have been an extensive amount of work anyway. We have however added further information to the Discussion section to more thoroughly compare our results with Matsubara et al., and more thoroughly describe the experimental differences. We hope that this will help to improve the manuscript despite our lack of additional experiments.

Reviewer 2 Report

The mutagenesis study is fine. You have obtained new results that suggest a novel mechanism in ssDNA binding and oligomerization of the Red-beta protein, by virtue of  the original assay methods.  The manuscript is readable and is arranged very well.  So, I recommend the editor to accept the manuscript as is.

Author Response

We thank the reviewer for their favorable remarks and support for publication as is.